# Intelligent Bearing Fault Diagnosis Based on Feature Fusion of One-Dimensional Dilated CNN and Multi-Domain Signal Processing

**DOI:** 10.3390/s23125607

**Published:** 2023-06-15

**Authors:** Kaitai Dong, Ashkan Lotfipoor

**Affiliations:** 1Mindsphere Analytics Centre, Digital Service, Siemens Mobility, London NW1 1AD, UK; 2Institute for Infrastructure and Environment, Heriot-Watt University, Edinburgh EH14 4AS, UK

**Keywords:** fault diagnosis, dilated convolution neural network, signal processing, feature extraction, anti-noise ability

## Abstract

Finding relevant features that can represent different types of faults under a noisy environment is the key to practical applications of intelligent fault diagnosis. However, high classification accuracy cannot be achieved with only a few simple empirical features, and advanced feature engineering and modelling necessitate extensive specialised knowledge, resulting in restricted widespread use. This paper has proposed a novel and efficient fusion method, named MD-1d-DCNN, that combines statistical features from multiple domains and adaptive features retrieved using a one-dimensional dilated convolutional neural network. Moreover, signal processing techniques are utilised to uncover statistical features and realise the general fault information. To offset the negative influence of noise in signals and achieve high accuracy of fault diagnosis in noisy settings, 1d-DCNN is adopted to extract more dispersed and intrinsic fault-associated features, while also preventing the model from overfitting. In the end, fault classification based on fusion features is accomplished by the usage of fully connected layers. Two bearing datasets containing varying amounts of noise are used to verify the effectiveness and robustness of the suggested approach. The experimental results demonstrate MD-1d-DCNN’s superior anti-noise capability. When compared to other benchmark models, the proposed method performs better at all noise levels.

## 1. Introduction

Rotating machinery is widely employed in mechanical systems and is critical in industrial applications. As one of the key components of rotating machinery, rolling bearings typically function in a complex and demanding setting and are hence prone to faults. Any problem with the rolling bearings may lead to the failure of the entire mechanical system, resulting in significant equipment downtime, severe financial losses, and even fatalities [1]. Therefore, it is vital to create effective intelligent fault diagnosis approaches to monitor the state of rolling bearings. Furthermore, recent technical advances have enabled the collection of massive amounts of monitoring sensor data from various parts of the equipment. Such information-rich sensor data has popularised and accelerated the usage and development of data-driven fault diagnosis tools. Over the past two decades, intelligent fault detection methods based on machine learning (ML) and deep learning (DL) have garnered widespread attention and are frequently deployed in real-world applications [2,3,4].

In practice, a typical intelligent fault diagnosis system employing traditional ML techniques consists of two important steps: (1) feature extraction and selection, and (2) fault classification. Signal inputs are processed to extract fault-sensitive information, which is then fed into ML models for fault recognition. Statistical feature information can be examined in time, frequency, and time-frequency domains through complex signal processing and analysis methods to uncover the operating conditions of bearings from various perspectives. In many circumstances, utilising features from multiple domains has been demonstrated to be more effective in distilling useful information than single-domain features for fault diagnosis problems [5]. Yan and Jia [6] employed various approaches to extract fault feature information from multi-domain aspects. It allowed for the raw vibration signals to be analysed for their inherent properties and the specifics of the bearing conditions. Abid et al. [7] defined a combined multi-domain feature set made up of time-domain and frequency-domain statistical features along with some special features. Multi-domain features derived from decomposed sub-band vibration signals were used by Ma and Wu [8] to enhance feature quality. Most recently, Yu et al. [9] conducted an experimental study on multi-domain fault features extraction for underwater vehicles, revealing effective improvement in feature identification. Furthermore, with the advanced development of artificial intelligence, deep learning has become increasingly popular in the field of fault diagnosis. Its deep architectures and nonlinear transformations enable it to capture representative information directly from input signals at multiple levels of abstraction [4]. Deep learning, in contrast to traditional ML methods, reduces the signal pre-processing steps and emphasises the mapping between signals and fault classes, thus minimising the need for empirical knowledge of signal processing. It has the ability to capture fault features from bearing signals in an adaptive manner and conduct the diagnostic process in an end-to-end way. Deep Convolutional Neural Networks (CNN) are the most adopted deep learning models in fault diagnosis applications, owing to their superior performance in hidden feature extraction and classification, particularly when large amounts of data are available [10,11]. Guo et al. [12] employed a novel hierarchical adaptive CNN to diagnose bearing faults and determine bearing severity. The model achieved 99.7% accuracy using data obtained from a test rig. Jing et al. [13] converted the original time-domain signal into the frequency spectrum and applied 2d-CNN to learn features and diagnose gearbox faults. It obtained noticeably better performance than manual feature extraction methods. In the works of Abdeljaber et al. [14], a 1d-CNN model with inherent adaptive designs to combine feature extraction and fault classification into a single block was adopted in the real-time damage detection system.

Bearing signals are often gathered in a very complex operation setting. Using a combination of features gleaned from multiple channels improves the ability to distinguish between different health conditions of rolling bearings. Several feature fusion algorithms have since been proposed to capitalise on the advantages of multiple fault diagnosis approaches or multiple sensory data sources [15]. Liang et al. [16] employed a parallel convolutional neural network (P-CNN) architecture to extract features in the time and time-frequency domains and fused multi-domain features to enrich fault information and improve model performance. Guo et al. [17] implemented a hybrid feature model that integrated nonlinear aircraft dynamics characteristics and DL-based features to identify fault signals from sensors on unmanned aerial vehicles (UAVs). Most recently, a bearing fault fusion method of incorporating empirical features and adaptive features extracted by a modified neural network structure (LiftingNet) was presented by Xie et al. [18] and it utilised XGBoost for the fault classification task. The result was validated by two motor bearings datasets and demonstrated improved accuracy.

Excellent diagnosis results have been produced in the past using ML or DL-based techniques [19]; however, these models have often been tested using bearing signals collected in a controlled laboratory setting, which is not representative of real-world industrial circumstances. To be more exact, the signal quality and subsequent feature learning capacity will suffer since real-world applications would inevitably include plentiful noise in collected vibration signals. Several CNN-based architectures with anti-noise capability have been proposed by researchers. For instance, to conduct bearing fault diagnosis under noisy environments and changing working loads, Zhang et al. [20,21] suggested an end-to-end CNN approach that greatly outperformed prior methods. Yao et al. [22] put forward a stacked inverted residual CNN (SIRCNN) to improve diagnostic efficiency while still retaining exceptional robustness and noise-fighting capabilities. Because the convolutional kernels used in these methods are typically adept at capturing local features, the inference of strong noise hinders the feature learning ability and leads to a certain decline in diagnosis performance.

The majority of the fault diagnosis techniques listed above run into the following issues: (1) despite the relatively good accuracy, the generalisation ability is not optimal when evaluating under different operating circumstances, and overfitting may occur; (2) in order to extract delicate information and achieve higher accuracy, the neural network architectures are unduly complicated, which further increases the computational complexity; (3) the performance of the model is hampered in noisy environments since the fault features are frequently highly coupled with noises and the local feature extractors could fail to detect relevant fault features.

As a result, this paper introduces a novel feature fusion architecture for rolling bearing fault diagnosis based on one-dimensional dilated CNN (1d-DCNN) and signal processing methods in multiple domains (time, frequency, and time-frequency domains). The model employs dilated CNN for its unique ability to expand receptive fields and capture more dispersed and intrinsic details without increasing computational cost [23,24]. These automatically derived features are then combined with the statistical features produced from the multi-domain signal processing stages. This allows the model to retain its robustness and generalisation abilities while taking advantage of both feature extraction techniques. Fully connected layers are used for the final fault classification task. Eventually, the model is trained in both a no-noise and Gaussian-noise setting. The primary contributions of this study can be summarised as follows:(1)A novel feature fusion model, MD-1d-DCNN, is built using multi-domain statistical characteristics and adaptive features from one-dimensional dilated CNN. It achieves greater robustness against noise than state-of-the-art benchmark approaches.(2)Bearing condition indicators that are reflective of bearing faults from several perspectives can be effectively evaluated utilising signal processing and analysis techniques in the time, frequency, time-frequency domains.(3)By introducing dilated CNN, we can learn features more effectively over an extended field and avoid getting stuck in local feature extraction. It aids in circumventing the overfitting issue and allows for the extraction of high-quality features in a noisy environment, expanding the scope of use for this fault diagnosis model.(4)The performance of the proposed approach is assessed by adopting two rolling bearing datasets created by the Bearing Data Centre at Case Western Reserve University and the Railway Technology Research Group of the Polytechnic University of Madrid, respectively. The experimental findings show that the suggested method provides exceptional fault diagnosis accuracy and anti-noise capabilities. It is indicative of strong performance in real-world situations.

## 2. Fundamentals

In this section, the essential concepts of wavelet packet transform (WPT) and one-dimensional dilated CNN, which are employed in the proposed method, are briefly described.

### 2.1. Wavelet Packet Transform

Wavelet packet transform, a technique for multi-scale time-frequency analysis, is an efficient tool for analysing nonlinear and nonstationary vibration signals [25,26]. The signal can be decomposed by WPT into multiple signal components with varied centre frequencies. As seen in Figure 1, WPT can break down the signal’s high-frequency component into greater detail than the traditional Wavelet Transform (WT) without omitting any information [27]. Consequently, it is a more effective instrument for analysing nonstationary vibration signals with high-frequency perturbations and substantial background noises.

The decomposition relationship in wavelet subspaces, denoted as $W_{j}^{n}$, can be described as $W_{j}^{n}=U_{j}^{n}=U_{j+1}^{2n}⨁U_{j+1}^{2n+1}$, where j∈Z. Figure 1 displays the schematic diagram of the wavelet packet structure decomposition at level 3, in which the top level of the WPT tree indicates the original time-domain signal. The wavelet packet coefficients resulting from this decomposition process can be formulated as follows:(1)$$c_{j+1,k}^{2n}=\underset{l\in Z}{\sum }c_{j,l}^{n}h_{0}\left(l-2k\right),\;\;\;\;k\in Z$$
(2)$$c_{j+1,k}^{2n+1}=\underset{l\in Z}{\sum }c_{j,l}^{n}g_{0}\left(l-2k\right),\;\;\;\;k\in Z$$
where l=1, 2, ⋯, λ, $n=0,\;1,\;\cdots ,\;2^{j}-1$, k is the translation factor, j is the layer number, and λ is the sampling length of the time series signal. Moreover, $c_{j+1,k}^{2n}$ and $c_{j+1,k}^{2n+1}$ are the wavelet packet coefficients, $h_{0}\left(\cdot \right)$ and $g_{0}\left(\cdot \right)$ are the low-pass and high-pass filters, respectively. After the wavelet packet coefficients are obtained through the decomposition process, the reconstruction operation is required to obtain the signal components at the various frequency bands. A three-layer wavelet packet decomposition results in eight signal components, as illustrated in Figure 1, with each component corresponding to its frequency band. As different fault types have different vibration frequencies, the corresponding changes in signals can be observed in different frequency bands, making it more effective in extracting relevant fault features.

### 2.2. One-Dimensional Dilated Convolutional Neural Network (1d-DCNN)

#### 2.2.1. One-Dimensional Convolutional Neural Network

Over the past decade, CNN’s capacity to learn complicated patterns and extract hidden features has made it the dominant tool for many machine learning tasks. Traditional deep CNNs have reached state-of-the-art performance in several applications such as image classification and recognition. Nonetheless, such deep CNNs are normally designed and constructed for two-dimensional (2d) signals. Input signals collected in bearing fault diagnostic tools are predominantly unidimensional (1d). Due to the mismatch between the kernel and signal dimensions, the direct utilisation of a deep CNN in a fault diagnosis system is not feasible. It necessitates a suitable 2d-to-1d transformation, which naturally increases the computational complexity. In response to this shortcoming, compact and adaptive 1d-CNNs have since been developed and implemented in a variety of practical 1d diagnostic applications [14,28]. Equation (3) describes mathematically the standard 1d convolution operation.
(3)$$x_{i}={\left(w*s\right)}_{i}=\overset{n}{\underset{f=1}{\sum }}w_{i+f}\cdot s_{i+f}+b$$
where $x_{i}$ is the output of the $i^{th}$ receptive field in the input, $w_{i+f}$ represents the 1d convolution kernel, $s_{i+f}$ is the element in the input sequence’s receptive field, and *b* indicates the scalar bias. The value n denotes the size of the receptive field, or kernel.

The output of the neuron at the hidden layer can be computed from the input $x_{i}$ using the activation unit. In this investigation, a non-linear activation function named the leaky rectified linear unit (Leaky ReLU) is used. Leaky ReLU function is an enhanced version of the ReLU activation function. ReLU activation function ($f\left(x\right)=\max\left(0,x\right)$) often suffers the ‘dying’ ReLU issue, i.e., the gradient will be set to 0 for all negative inputs, meaning the corresponding neurons in the network will remain inactive regardless of the inputs supplied. Leaky ReLU, as represented in Equation (4), is designed to address this issue by returning small linear component of x for negative inputs. The slope coefficient α is determined prior to training.
(4)$$f\left(x\right)=\max\left(\alpha x,x\right)$$

Batch Normalization (BN) is another important operation, which is used to standardise and normalise the network’s batch inputs. It helps speed up and stabilise the training process of the deep neural network, address the internal covariate shift issue, and improve the robustness of the network. Mathematically speaking, the BN process first subtracts the mean value (μ) of the batch inputs ($y_{i}$) and then divide it by the sum of the batch’s standard deviation (σ) and the smoothing term (ε), as seen in Equation (5). The smoothing term is employed to prevent a division by a zero value. Furthermore, two additional trainable parameters, i.e., re-scaling (λ) and shifting (β) parameters, are introduced at the end in Equation (6) since scaling inputs by a randomly initialised parameter decreases the accuracy of the weights in the subsequent layer. This ensures that the optimal values for two parameters are chosen, allowing for accurate batch normalisation.
(5)$$\hat{y_{i}}=\frac{y_{i}-\mu }{\sigma +\varepsilon }$$
(6)$$z_{i}=\lambda \cdot \hat{y_{i}}+\beta$$

The pooling layer, often referred to as the down-sampling layer, is usually added behind the 1d convolution process to decrease the dimension of feature maps and the number of related parameters while preserving the most prominent features. Max pooling is an aggregating procedure in which the maximum value is extracted using a window with a scalable size that slides over the feature map with a pre-defined stride of length. Within the region R on the feature map, max pooling can be summed up in the following formula:(7)$$z=\underset{i\in R}{\max}z_{i}$$

Given the fact that the size of the convolution kernel is typically kept low in traditional CNNs, convolution kernels can only cover limited receptive fields, allowing it to extract just local information. The number of parameters and computational cost will both rise when the size of the convolution kernel increases. As a result, dilated convolutions are introduced to make up for this drawback of standard convolutions.

#### 2.2.2. Dilated Convolution

When compared to standard convolutions, dilated convolutions are able to learn and aggregate features over a wider time window [29]. By gathering additional feature information without raising the overall number of trainable parameters or rearranging the sequence of the input data, it provides benefit that standard convolutions lack. Therefore, convolutions with dilated filters are frequently employed to enhance feature extraction ability over an extended time horizon [28,30]. The dilation structure has displayed exceptional capabilities in a diverse range of time-series analyses and image denoising applications [28,31,32]. In this work, we adopt 1d dilated convolutional neural networks (1d-DCNN) to capture sensitive fault information from rolling bearing signals.

The receptive field size of the convolution kernel is what primarily differentiates standard convolution from dilated convolution. As can be seen in Figure 2, a convolution with a dilation factor of 1 (d=1) is identical to a standard convolution, whereas a convolution with a dilation factor of greater than 1 (d>1, d∈ℤ) expands the receptive field by skipping pixels between consecutive elements. In principle, it is equivalent to inserting “0” between the elements of the convolution kernel. For a 1∗k convolution kernel, the size of the dilated convolution kernel $k_{d}$ can be defined using Equation (8):(8)$$k_{d}=k+\left(k-1\right)\left(d-1\right)$$

By having a fixed gap (d−1) between elements, which are depicted in Figure 2 as coloured squares, the dilation operation enlarges the receptive field of the convolution kernel and covers a larger amount of the input information. Therefore, when the dilation factor d is set to 2, the 1∗3 convolution kernel is expanded to a 1∗5 dilated convolution kernel. Similarly, it will be expanded to a 1∗7 dilated convolution kernel when d=3, so on for increasing values of d. In this model, both the structure of the dilation and the dilation factor for each convolution layer are fixed.

Moreover, in dilated convolution the receptive field can be enlarged exponentially by stacking layers of convolutions with increasingly dilated values. Figure 3 provides an intuitive visual comparison between traditional convolution and dilated convolution with a three-layer convolution structure. In the output layer, the tradition CNN can only capture four inputs in the signal. With adjusting the layer depth or the kernel size, dilated CNN can acquire 8 data points under the same conditions. This demonstrates that the dilated CNN can learn more essential knowledge of the wider context without any loss of resolution.

Furthermore, let d be a dilation factor and ${*}_{d}$ denote the dilated convolution and Equation (3) can be rewritten to define the 1d dilated convolution operation [28], as shown in Equation (9).
(9)$$x_{i}={\left(w{*}_{d}s\right)}_{i}=\overset{n}{\underset{f=1}{\sum }}w_{i+d*f}\cdot s_{i+d*f}+b$$

## 3. The Proposed Method

### 3.1. Model Structure

The proposed bearing fault diagnosis model is based on the examination of calculated statistical characteristics describing the health conditions of machines and the adaptive features distilled from the deep dilated convolutional neural network. Figure 4 depicts the suggested method’s data processing workflow. The model structure is made up of two essential modules: the feature extraction module and the fault classification module. The feature extraction module is composed of two parallel components. The first component contains classic statistical features extracted from bearing vibration signals in time, frequency, and time-frequency domains utilising expertise and prior knowledge. These representative fault characteristics have been proven to be useful and effective in detecting and quantifying relevant signal parameters [1,4]. The other component uses a one-dimensional dilated CNN (1d-DCNN) model to extract nonlinear and adaptive features from the bearing signal. Additionally, spectrum data can significantly shorten the sequence and improve the information representation during the feature extraction stage. Therefore, before transferring the vibration signal into the 1d-DCNN model, fast Fourier Transformation (FFT) was applied. 1d-DCNN structure allows for the learning of intricate links between signals and corresponding fault classes as well as the acquisition of sophisticated hidden features. High-dimensional features derived from two parallel components are likely to possess correlative and redundant information. Simply utilising these features as input to a classification module will result in poor diagnostic performance. Therefore, 16 representative features from each parallel component are combined as the input to the classifier.

### 3.2. Feature Extraction

#### 3.2.1. Data Preprocessing and Sequence Generation

The original bearing vibration data are essentially one-dimensional time series. These unprocessed data have a high degree of noise and variability and come in different lengths. A standard data preprocessing step is presented to normalise the vibration data and produce smaller samples of data from the original time series. The vibration data can be normalised to fall within the range of [0, 1], using the Min-Max normalisation method [18]. The calculation formula is given in Equation (10):(10)$$x^{*}=\frac{x-x_{min}}{x_{max}-x_{min}}\times \left(c_{max}-c_{min}\right)+c_{min}$$
where $x_{max}$ and $x_{min}$ represent the maximum and minimum values of the original measurement signal, respectively, and $c_{max}$ and $c_{min}$ are the maximum and minimum normalisation range. Both the training and test data sets will be subjected to this normalisation process.

The sequence generation phase can be thought of as the development of smaller, non-overlapping sub-samples from the original time series. Usually, the number of data points N in a complete rotating motor shaft is calculated by $N=f_{s}\cdot 60/\omega$, where $f_{s}$ represents the sampling frequency and ω is the rotating shaft speed in rpm. The sample length should be larger than the computed value to ensure each of the smaller samples contains at least a full bearing vibration cycle of the recorded data.

#### 3.2.2. Multi-Domain-Based Fault Feature Extraction

Signal processing techniques in the time, frequency, and time-frequency domains are used to uncover fault characteristics in vibration bearing signals. The term “time-domain signal” typically refers to the initial vibration signal that was recorded from the rotating machinery. When a fault develops, the mechanical structures in its vicinity are altered, generating an impulse or shock that causes the variation in vibration signals. Additionally, these time-domain signals’ amplitudes and distributions may alter. As a result, time-domain statistical features are used to characterise the mechanical operating conditions based on vibration signal waveforms. For instance, the trend, magnitude, and energy of a time-domain signal can be reflected in several metrics, such as the root amplitude, root mean square, and peak value. In most cases, a fault can cause irregular mechanical vibration and elevate these metrics, which can clearly indicate how severe a fault is when it becomes worse. Nonetheless, they are insensitive to weak incipient faults. Because of this, additional features are used to describe the time series distribution and gauge the impulse of signals, including kurtosis value, crest factor, clearance factor, shape factor, impulse factor and so on. These features are robust to changing operational environments and are reliable early warning signs of faults. As indicated in Table 1, a total of 8 time-domain features are chosen for this analysis. While time-domain features can display certain underlying characters of a signal, they are not sufficiently strong enough to reveal the underlying character of a signal and they cannot accurately portray the mechanical health issues of rotating machinery. One effective and potent tool for analysing stationary signals is frequency spectrum analysis, which can identify the frequency distributions and components of gathered vibration signals. The Discrete Fourier Transform (DFT), which is efficiently computed by FFT, is used to construct a frequency-domain spectral representation of signals. However, the majority of bearing vibration signals are sporadic or nonstationary. This scenario normally prevents the functional use of the conventional Fourier transform. Nevertheless, the vibration signals can still be regarded as roughly stationary for finite data lengths [1]. As a result, 8 sensitive frequency-domain statistical features can be utilised to represent the bearing operational state from the frequency spectrum to some extent. The formulas for these adopted frequency-domain features, which are grouped as indicators of frequency position change and indicators of energy, are listed in Table 1.

In order to obtain more characteristic fault information, WPT is utilized to generate additional sub-signals, allowing for the extraction of more advanced features from nonstationary vibration signals. As displayed in Figure 5, the vibration signals are decomposed by db5 WPT at level 3, producing $2^{3}=8$ frequency-band signals. 8 statistical features in the time domain are obtained from each of the 8 frequency-band signals in a manner similar to the feature extraction method described above. This results in additional 64 features. Moreover, demodulation is introduced to reduce the impact of extraneous information on fault identification. As a result, the decomposed signals are further demodulated using the Hilbert transformation due to its superior capability to detect incipient bearing faults even in the presence of heavy background noise [33]. Another feature set including 64 frequency-domain features is subsequently derived from the envelop spectrum of each decomposed signal. Finally, by virtue of WPT, 9 time-frequency-domain features that reflect the energy distribution of the frequency-band signals are also acquired. The energy and energy entropy of these decomposed signals in the independent frequency channels, as shown in Table 1, carry a wealth of information about operational states and faults of certain mechanical parts, which can facilitate the efficient monitoring and diagnosis of various mechanical issues [34]. In summary, the multi-domain signal processing steps discover a total of 153 fault-related features. These complementary features uncover fault characteristics from multiple perspectives and have been shown to be highly effective and ubiquitous in a variety of fault diagnostic tasks [1,7].

Despite the fact that the aforementioned extracted features may disclose rolling bearing faults from a variety of angles, not all of them are sensitive or closely associated to the fault conditions. To enhance classifier performance and avoid the dimensionality curse, it is crucial to extract the information that is most directly connected to faults and toss out irrelevant or redundant features. Therefore, the detected features are fed into a three-layer dense layers to reduce the size of learned representations down to 16. The characteristics from the output will then be integrated with the outcomes from the other parallel feature extraction component.

#### 3.2.3. 1d-DCNN-Based Fault Feature Extraction

As displayed in Figure 4, fast Fourier transform is first applied to the time-domain vibration signal to enhance information representation. The transformed data will go through the 1d-DCNN architecture that consists of four dilated convolutional blocks in parallel-stack form. Each dilated convolution block is a stack of one-dimensional dilated convolution layer, batch normalisation, activation layer, and max-pooling layer. The parameters employed in each block are specified in Table 2. The stacked dilated convolution blocks are effective in expanding the receptive field and capturing more signal context information without increasing computational cost. Dilated convolution is followed by Leaky ReLU to help propagate gradients efficiently and create a feature map. Dilated CNN computation is already shown in the Section 2.2 with great details. In this analysis, 4 layers of dilated convolutions of increasing dilation width of 1, 2, 4, and 4 are utilised to maximise the feature extraction performance in noisy settings. After flattening the output of dilated convolutional layers, the information is passed to a fully connected layer to obtain the engineered features. A dropout layer is added to random discard some dense layer weights during training to reduce overfitting.

### 3.3. Feature Fusion and Loss Function

The purpose of feature fusion is to combine feature information from different sources and further decrease the feature dimensions to reduce the classification difficulty. In this case, the statistical features extracted from multiple domains are fused with high level representations from one-dimensional dilated convolutional operations to enhance the fault feature. The statistical features chosen for the task are well recognised fault indicators that have been tested and applied to appropriately depict diverse fault conditions. The extraction and utilisation of these general features ensure the robustness of the fault classification model. As a result, 153 fault-related features are produced by the multi-domain feature extraction module. Following that, three dense layers are adopted to reduce the dimension and remove the redundant information from the original statistical features, yielding a total of 16 features. The feature number is the same as that of the adaptive features, which are obtained from the last fully connected layer output in the adaptive feature extraction module. These two complementary feature sets are fused into a concatenated feature vector of dimension 32, and then used as the input to the fully connected layers to learn the feature interactions and dependencies and map the distributed feature representation to the label space. This allows for a full merging of both general characteristics and more nuanced, hidden features. Two consecutive full connection layers are employed in the end to improve the non-linear fitting ability of the network and the weights in each layer are randomly initialized and trained for optimization. In order to measure the quality of fusion features, the softmax function (σ), as shown in Equation, is used to calculate the posterior probability distribution of the target.
(11)$$\sigma \left(z_{i}\right)=\frac{e^{z_{i}}}{\sum_{j=1}^{M}e^{z_{j}}}$$
where $z_{i}$ is the $i^{th}$ input feature of the softmax function and M is the number of categories in the multi-class classifier. Moreover, the loss function calculates the error between the predicted value and the true value, backpropagates the error from the last layer to each layer of the network, and updates the weights. The updated parameters continue to participate in the training, looping back and forth until the loss function value reaches the minimum; that is, the goal of the final training is reached. In this paper, cross-entropy loss function is adopted, denoted as $L_{c}$, and it can be computed using Equation.
(12)$$L_{c}=-\frac{1}{N}\overset{N}{\underset{i=1}{\sum }}\overset{M}{\underset{j=1}{\sum }}p_{i,j}\log\left(q_{i,j}\right)$$
where N is the total number of samples, $q_{i,j}$ is the predicted probability of observation i belonging to class j, and $p_{i,j}$ is the sign function (0 or 1) that represents the label value of the observation. To enhance the robustness of features and prevent overfitting, regularization term is added to the loss function L.
(13)$$L=L_{c}+\lambda L_{r}$$
(14)$$L_{r}=\underset{\theta }{\sum }\theta^{2}$$
where λ is the regularization coefficient, θ is the network parameter, and the term $L_{r}$ in Equation (14) is L2 regularization term and it can improve the sparseness of the network.

## 4. Experimental Validation

Extensive experiments have been carried out on two different rolling bearing datasets to assess the performance of the proposed method. In Case 1, the benchmark bearing dataset provided by the Case Western Reserve University (CWRU) Bearing Data Centre [35] is used to validate the suggested model. The dataset used in Case 2 was captured by the Railway Technology Research Group (Centro de Investigación en Tecnología Ferroviaria—CITEF) at the Polytechnic University of Madrid utilising a low-cost, high-frequency data acquisition device [36,37]. All validations are conducted on Google Colab platform with an Intel Xeon processor and Tesla P100 GPU with 26 GB of RAM. The suggested model is written and developed using Keras’s functional API with TensorFlow 2.1.0 backend in Python 3.6.

### 4.1. Case One: The CWRU Bearing Data

#### 4.1.1. Experiment Setup and Data Description

The CWRU Bearing Data Centre’s rolling bearing dataset is frequently used as a benchmark dataset to assess the effectiveness of various intelligent fault diagnosis methods. Figure 6 depicts the test rig that was used to acquire the dataset. It primarily comprises of a 2-hp Reliance Electric motor, a torque transducer/encoder, accelerometer, control electronics, and a dynamometer. The deep-groove ball bearing (6205-2RS JEM) manufactured by SKF served as the tested bearing. Under motor loads of 0 hp, 1 hp, 2 hp, and 3 hp (1 hp = 746 W), vibration data recorded at the drive end of the motor housing with a sampling frequency of 12 kHz were utilised in this investigation. Single point faults were introduced by electro-discharge-machining (EDM) with fault diameters of 0.007, 0.014, and 0.021 inches (1 inch = 25.4 mm) at the inner race (IR), the rolling element (RA), and the outer race (OR) of the bearing. In addition, 0.028 inches damage was also introduced on the IR and RA. For each motor load, the healthy bearing condition was also added in addition to the aforementioned bearing fault types. Therefore, a total of 12 bearing health condition types are included in this analysis, as shown in Table 3.

We take 1024 data points of vibration signal sequence as a sample and obtain 120 samples for each bearing condition. Therefore, a total number of 1440 samples are included in the dataset. We split the dataset into train, validation, and test sets in a ratio of 70%, 20%, and 10%, respectively.

#### 4.1.2. Model Performance Metrics

Accuracy is adopted as the main evaluation metric in this study. In the realm of fault diagnosis, it is one of the most widely utilised performance indicators. Equation (15) provides a mathematical definition for accuracy.
(15)$$\mathrm{accuracy}=\frac{\mathrm{TP}+\mathrm{TN}}{\mathrm{TP}+\mathrm{TN}+\mathrm{FN}+\mathrm{FP}}$$
where the terms TP (true positive) and FN (false negative) refer to the number of actual positive examples that are correctly classified as positive and those that are incorrectly identified as positive samples, respectively. Similarly, the term TN (true negative) represents the number of actual negative examples correctly classified as negative, while FP (false positive) is the number of actual negative samples incorrectly classified as positive. In most cases, a higher accuracy reflects an improved classifier performance.

In addition, confusion matrix is also used to summarise and visualise the performance of multi-class classifiers in this paper. A confusion matrix displays the proportion of correctly classified cases as well as the proportion of cases that are misclassified among various categories, i.e., the values along the diagonal represent the proportion of correct classifications made by the algorithm, while the other numbers represent the errors made.

#### 4.1.3. Model Evaluation

In this section, the MD-1d-DCNN model is assessed by means of CWRU bearing dataset. Several ML/DL and fusion models for fault diagnosis, including random forest (RF), support vector machine (SVM), one-dimensional dilated convolutional neural network (1d-DCNN), CNN-LSTM (long short-term memory), and XGBoost-fusion (empirical features fused with adaptive features from LiftingNet) [18], denoted as XGBF for simplicity, are employed to evaluate the efficacy and practicality of the suggested method. It is important to note that the curse of dimensionality might arise when using SVM on raw data without carrying out feature reduction, which is why principal component analysis (PCA) is frequently paired with SVM to boost the classification performance. This method will henceforth be referred to as PCA-SVM throughout the remainder of the paper. Moreover, each model will be tested five times and the average accuracy of five tries will be adopted as the evaluation result.

As seen in Table 4, the proposed MD-1d-DCNN achieves the highest testing accuracy of 100%, outperforming the XGBoost-based fusion model by 0.3% and by at least 4.5% compared to traditional ML models such as PCA-SVM and Random Forest. By utilising a set of confusion matrices, the diagnosis outcome is graphically presented against a variety of fault labels for each model in Figure 7. The suggested method exhibits outstanding classification performance across all fault types, whereas other benchmark models have missed predictions for a variety of labels. Moreover, Table 4 displays the training duration for each model used in this experiment. DCNN-based models are noticeably quicker than those built with other approaches. Due to the sequential computation in the LSTM layer, the training time for the CNN-LSTM model is lengthy. The findings show that the proposed method yields desirable outcomes in terms of accuracy and training duration. In addition, it is necessary to point out that the CWRU dataset’s bearing fault signals are representative of the standard public dataset given that it was collected using high-quality equipment in controlled experimental settings. In most cases, it is straightforward to determine the fault issue when examining the CWRU vibration data; hence, the performances of all models on the CWRU data are generally satisfactory, even when inspecting the classification results in finer detail for each fault label.

#### 4.1.4. Model Performance under Various Noise Levels

Conventional laboratory bearing testing employs single-point damaged rolling bearings in a quiet setting. However, the efficacy and adaptability of fault diagnosis methods are frequently hindered by strong noises interferences in the bearing signals gathered from an operational environment. In this section, the proposed model will be evaluated on the CWRU dataset under various noise levels. As a result, a Gaussian white noise is added to the raw vibration signals to imitate the data acquired in a noisy industrial environment. This is done so that the performance of the model mirrors the real-world scenario more closely. The signal-to-noise ratio (*SNR*), as given by Equation (16), is commonly employed to quantify the noise level.
(16)$$SNR=10\;\log_{10}\left(\frac{P_{signal}}{P_{noise}}\right)$$
where $P_{signal}$ and $P_{noise}$ represent the power of signal and Gaussian noise, respectively. Figure 8 displays the time signal and time-frequency spectrum of the 0.007-inch-fault bearing signal with five different amounts of additional noise, ranging from −6 dB to 6 dB, in order to better highlight the attributes of noisy signals in various domains. The spectrum is produced using short-time Fourier transform (STFT) over time windows [38], where the time outputs correspond to time window centres. As shown in Figure 8, the spectrogram is clearly capable of revealing certain fault features, which are difficult to differentiate from jumbled time signals, and collecting vital visual information regarding bearing fault. More importantly, when the noise grows, it dilutes the fault characteristics, making it harder to identify frequencies that are associated with bearing faults. As the SNR approaches −3 dB and −6 dB, it is difficult to visually determine the fault characteristics. Consequently, it significantly magnifies the difficulty of fault diagnosis and can be served as a good way to test the model’s anti-noise capability.

In this experiment, we introduced nine distinct noise levels ranging from 10 dB to −6 dB (from low to high) into the raw bearing data. The suggested MD-1d-DCNN model’s testing accuracy is compared to that of RF, PCA-SVM, 1d-DCNN, CNN-LSTM, and XGBF. Figure 9 depicts the outcomes of the analysis. It is evident that the MD-1d-DCNN model gets the best diagnostic performance compared to other benchmark models in all noise settings. At an SNR of larger than 0 dB, the MD-1d-DCNN model tests with near-perfect accuracy; while at an SNR of −6 dB, the accuracy drops slightly but still exceeds 96%. When the degree of noise is low (the SNR is high), on the other hand, all the benchmark models have performed similarly well, scoring between 90.5% and 99.8%, indicating that low-degree noise only has a small impact on the diagnosis. In other words, the benchmark models’ limited feature detection capability is most likely to blame for the diagnostic errors. Yet, when noise increases, benchmark model performance deteriorates. When the SNR values are less than 0, the XGBoost-based fusion model delivers relatively good classification performance, with accuracies between 98.6% and 99.8%, only lagging behind the MD-1d-DCNN by 1.4%. However, when SNR falls below 0, diagnostic accuracy significantly decreases. In a situation when the signal-to-noise ratio (SNR) is −6, the suggested MD-1d-DCNN performs 8.3% better than the XGBF. In comparison, other benchmark models suffer more drastic performance decline, with as much as a 30% gap between different SNRs, e.g., PCA-SVM’s accuracy at an SNR of 10 dB is 93.1% but the value decreases to 63% when SNR moves down to −6 dB. After SNR falls below 2 dB, this trend becomes glaringly obvious. Even at the greatest noise level (SNR = −6 dB), where the CNN-LSTM model excels among other benchmark methods with a diagnostic accuracy of 78.5%, it is about 18% less effective than the suggested MD-1d-DCNN. It highlights the advantage of the MD-1d-DCNN over other approaches in feature learning and recognition in a highly noisy setting. This is primarily attributable to the following two factors: First, the model can extract features from a variety of facets, which helps to counteract the negative effect of noise on fault-relevant feature detection inside only a single domain. Second, noise can contaminate fault information and harm feature learning in traditional CNN-based models, which focus on local features as the kernel slides across the signal. The MD-1d-DCNN, however, applies dilated filters to acquire and aggregate features over an extended time span, thus making it more robust against ambient noise. Due to its excellent anti-noise capabilities and avoidance of additional denoising processing, it is therefore more suited for bearing fault diagnosis in actual operation.

### 4.2. Case Two: The CITEF Bearing Data

#### 4.2.1. Experimental Setup and Data Description

In this case study, the bearing dataset created by the Railway Technology Research Group (CITEF) at the Polytechnic University of Madrid is used to test the proposed model. The bearing test bench setup, as shown in the Figure 10, was utilised to obtain the bearing signals. It employs a series BL 110 synchronous electric servomotor to generate traction. The equipment enables for the simultaneous testing of three sets of roller bearings, two of which are 22205E1KC3 double-row spherical roller bearing housings, supported by the SN-505 casing. Moreover, three SKF 6304-2R ball bearings (middle) are placed continually between the housings. Accelerometers are positioned in the centre of the bearing casings and clamping tower. The orientation of Accelerometer 2 is upward, with Accelerometers 1 and 3 pointing down. This means it can detect even the smallest variations in the vibratory bearing’s behaviour [39]. Despite having recordings from all three accelerometers, only the vibration data from Accelerometer 1, labelled as ‘Rod_1′ in Figure 10, is included in the CITEF dataset.

Bearing signals in both healthy and faulty states are included in the CITEF dataset. Four fault levels resulting from localised defects in the outer race (OR) and rolling element (RE), under three operating conditions (200 rpm, 350 rpm, and 500 rpm), make up the faulty state signals. The damage descriptions are listed in Table 5. By repeating each run three times, it ensures that any variances in results due to environmental factors like temperature and humidity were distributed evenly across all data. As a result, the dataset contains 45 bearing vibration records. Each record 30 seconds’ worth of data, sampled at 40 kHz under a load of 1.4 kN. Based on the similar sequence generation rule defined in Section 3.2.1, we take 14,000 data points of vibration signal sequence as a sample and hence obtain 765 samples for each fault condition. There are a total number of 3825 samples in the dataset. The following phase involves splitting these samples into train, validation, and test sets for our model.

In addition, five different types of raw bearing signals and their corresponding time-frequency spectrograms, calculated using consecutive Fourier transforms, are displayed in Figure 11 for the reader’s perusal and comprehension. It is evident that different combined fault conditions produce different vibration signal signatures, i.e., different frequency distributions, and each frequency also possesses a unique time distribution. Such signal characteristics would be beneficial during the fault diagnosis stage.

#### 4.2.2. Model Evaluation

The CITEF combined fault dataset results for each model are compared in Table 6. Desipte the fact that the combined fault classes in CITEF dataset are not as easy to identify as in the CWRU examples, our MD-1d-DCNN still achieves a testing accuracy of 99.35%. Nonetheless, other benchmark models’ classification accuracies have dropped to a lesser extent. For instance, when compared to the corresponding CWRU test result, the performance of 1d-DCNN has decreased by 1.9%, while PCA-SVM’s result has fallen from 95% to 91%, representing a siginificant deterioration of 4%. Moreover, the results of the experiment are consistent with Table 4’s depiction of training duration. The suggested method’s training time is second fastest to that of the 1d-DCNN model, thanks to the usage of a parallel feature extraction component. When compared to the next quickest model, i.e., the PCA-SVM, the MD-1d-DCNN is still 2.6 times faster. In addition, Figure 12 compares the confusion matrices of the proposed method and the other benchmark methods. It exhibits the excellent performance of MD-1d-DCNN with all fault labels, even the less evident ones. Other methods have incorrectly classified bearing faults in various cases. This is owing to the fact that MD-1d-DCNN’s comprehensive feature engineering and extraction steps assist exploit information about the underlying issue of the rolling bearings, hence facilitating the identification of the most discriminating charactersitics in signals from a wide variety of angles. The MD-1d-DCNN model benefits greatly from these extracted statistical and adaptive features. It highlights, once more, the effectiveness of the proposed paradigm for fault diagnosis.

#### 4.2.3. Model Performance under Various Noise Levels

In this section, the effectiveness of our proposed method under noisy conditions is assessed. Due to the delicate nature of the bearing signals collected in the aforementioned environment, the SNR values between 0 and 10 dB are considered in this analysis. Again, as demonstrated in Figure 13, when it comes to signal classification with noise, our proposed MD-1d-DCNN model clearly excels above the benchmark methods. When the SNR is between 6 and 10, the XGBoost-based fusion model is quite close to MD-1d-DCNN. However, as the noise is increased, the performance disparity widens, with the accuracy difference increasing from 2.5% to 6.2%, demonstrating stronger anti-noise capabilities of the suggested technique. On the other hand, 1d-DCNN and CNN-LSTM achieve satisfactory accuracy and consistency when the added noise is minimal. However, both CNN-based models have experienced more significant loss of classification accuracy as the noise level rises than PCA-SVM. As the SNR value decreases from 10 dB to 0 dB, the accuracy of 1d-DCNN falls from 94.7% to 71.7%, or by 23%, while the accuracy of CNN-LSTM declines by 27%. In contrast, PCA-SVM appears to be somewhat more resilient and robust to growing noise levels, with a 16% loss in diagnostic accuracy, when the noise level goes from 10 dB to 0 dB. The suggested feature fusion method is validated by comparison findings showing improved classification accuracy when the fault type is obscure. Combining deep learning with a small amount of past knowledge has demonstrated promising outcomes in engineering practice and may even help non-professionals make more informed decisions. Eventually, MD-1d-DCNN achieves an accuracy of 85.8% at an SNR of 0 dB, whereas the best performance from other approaches reaches only about 79.6% accuracy, validating superior fault identification and adaptive feature learning under noisy environments. The results shown in Figure 13 are in line with what concluded from Case 1, exhibiting the same anti-noise capability and promising potential for practical use.

## 5. Conclusions

In this work, we present a feature fusion model called MD-1d-DCNN. Using signal processing methods, we first extract time, frequency, and time-frequency information from the vibration signal that is attributed to the fault mechanism. A 1d-DCNN model is then developed to improve feature extraction capability and prevent overfitting. Statistical features from multiple domains are combined with adaptive features from 1d-DCNN to boost anti-noise performance while maintaining the model’s generalisation potential. Two rolling bearing datasets are used to validate the classification model’s accuracy and robustness. The MD-1d-DCNN model has achieved 100% and 99.35% accuracy for CWRU and CITEF bearing datasets, respectively. Under a noisy environment, the proposed MD-1d-DCNN model has been shown to be effective and feasible in fault diagnosis, outperforming other conventional methods under highest levels of noises by at least 8.3% and 6.2% on CWRU and CITEF datasets, respectively. Its strong anti-noise capability bodes well for practical use.

**Limitations and future research directions:** First of all, the suggested method was initially tested to laboratory data that included Gaussian white noises. It has certain level of resemblance to actual noise characteristics, but it cannot fully represent the complexity of noisy operational environment. It would be beneficial to collect data in a real-world scenario to further verify the effectiveness of the method. Secondly, when compared to other DL-based models, the proposed algorithm exhibits promising training speed. Validating its inference efficacy and accuracy in a production setting will help get it closer to deployment. Lastly, this study achieves effective diagnostic findings with laboratory data collected in a stable working environment, using supervised learning. However, in real-world industrial applications, acquiring adequate well-labelled data is a major challenge. We plan to evaluate and enhance the suggested method’s performance in the context of semi-supervised learning. Moreover, further research into novel generative adversarial network (GAN) approaches for imbalanced small datasets is also planned in the near future.

## Figures and Tables

**Figure 1 sensors-23-05607-f001:**
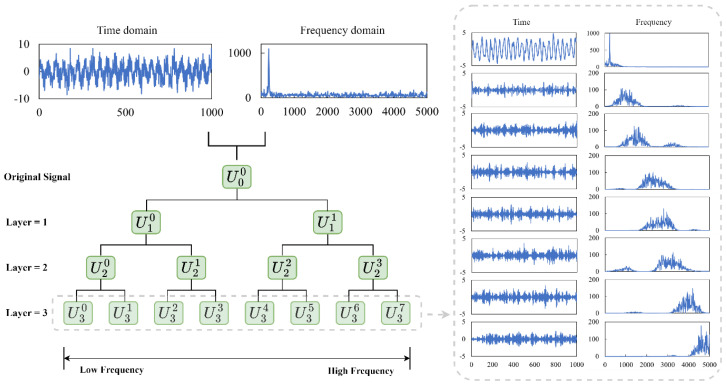
Wavelet packet transform (WPT) decomposition tree structure with layer *j* = 3 and signal details in time and frequency domains.

**Figure 2 sensors-23-05607-f002:**
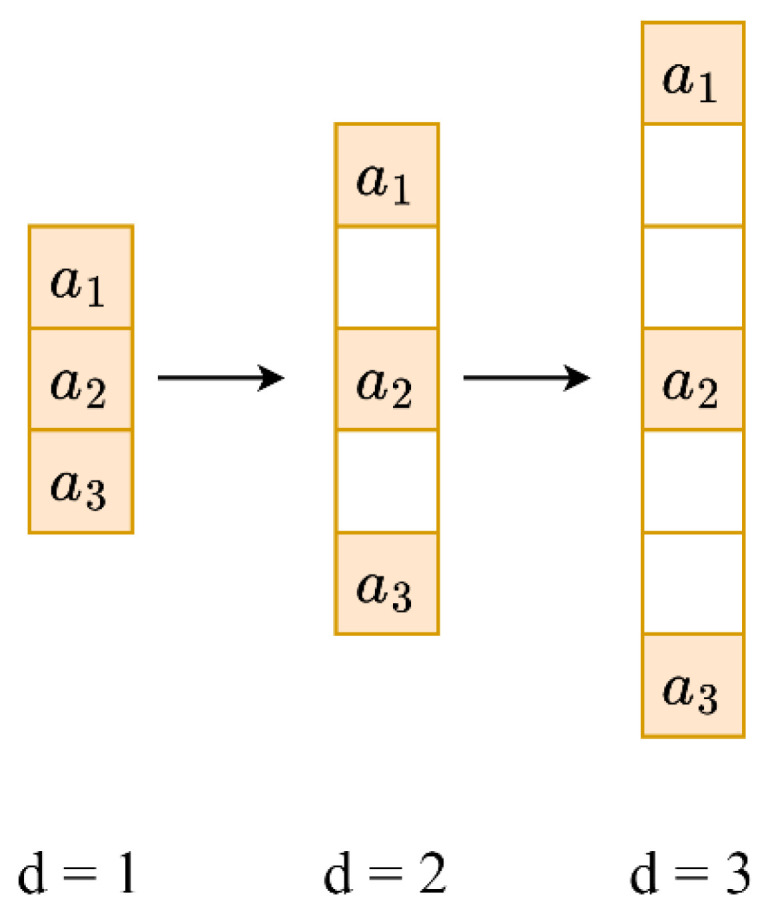
Dilation kernels with various dilation rates (when *d* = 1, it is equivalent to the standard convolution operation).

**Figure 3 sensors-23-05607-f003:**
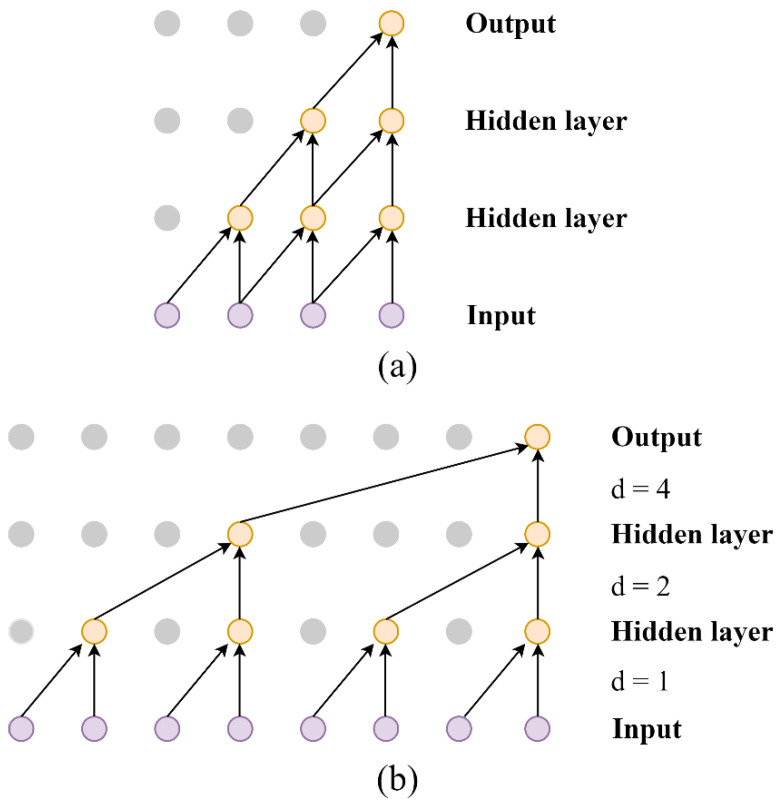
Schematic diagram of comparison between traditional convolution and dilated convolution with a three-layer structure. (**a**) Traditional convolution. (**b**) Dilated convolution.

**Figure 4 sensors-23-05607-f004:**
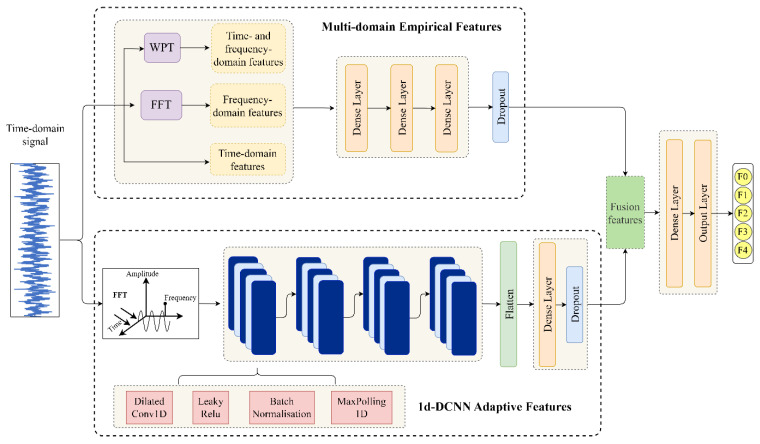
Network architecture of the proposed bearing diagnosis model.

**Figure 5 sensors-23-05607-f005:**
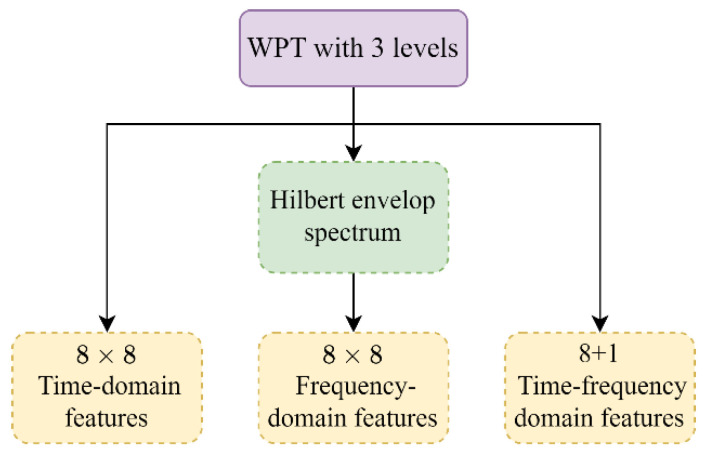
Feature extraction from multiple signal domains using WPT.

**Figure 6 sensors-23-05607-f006:**
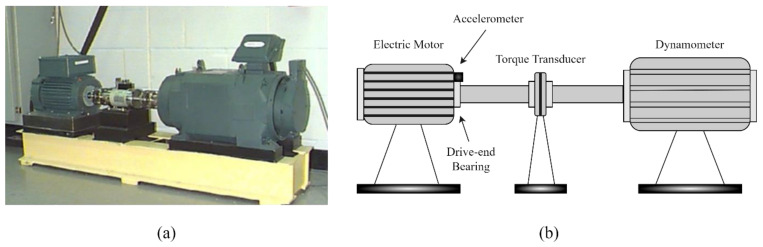
(**a**) The CWRU bearing fault test rig; (**b**) Schematic diagram of the CWRU experimental setup.

**Figure 7 sensors-23-05607-f007:**
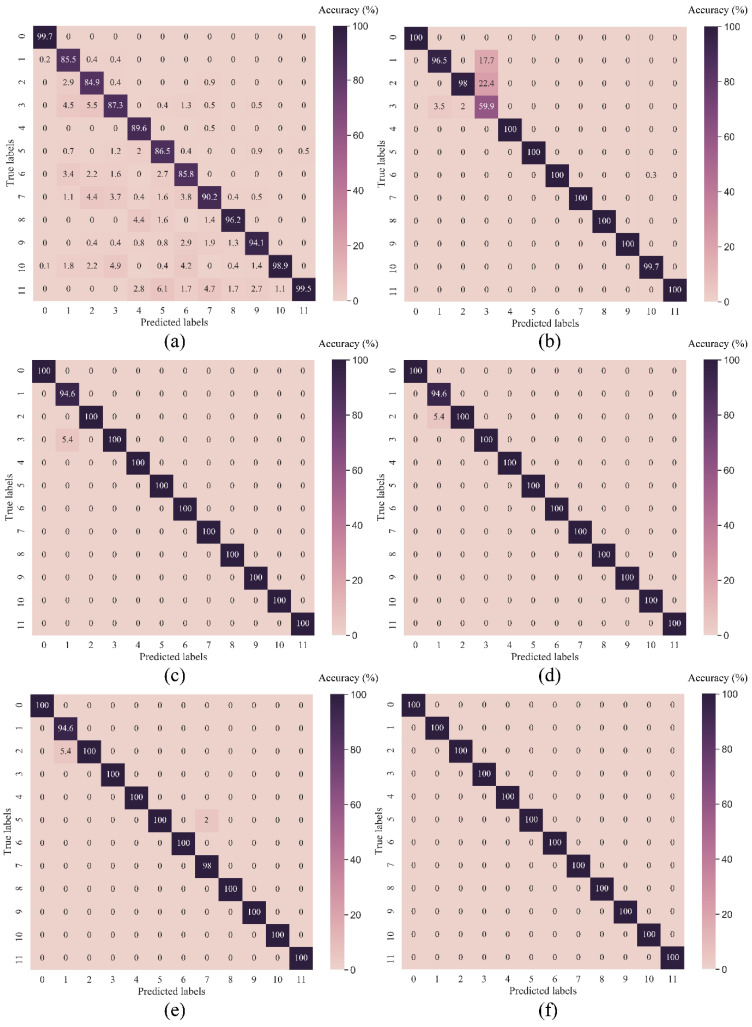
Confusion matrices of various methods in testing samples of CITEF dataset (**a**) RF; (**b**) PCA-SVM; (**c**) 1d-DCNN; (**d**) CNN-LSTM; (**e**) XGBF; (**f**) MD-1d-DCNN.

**Figure 8 sensors-23-05607-f008:**
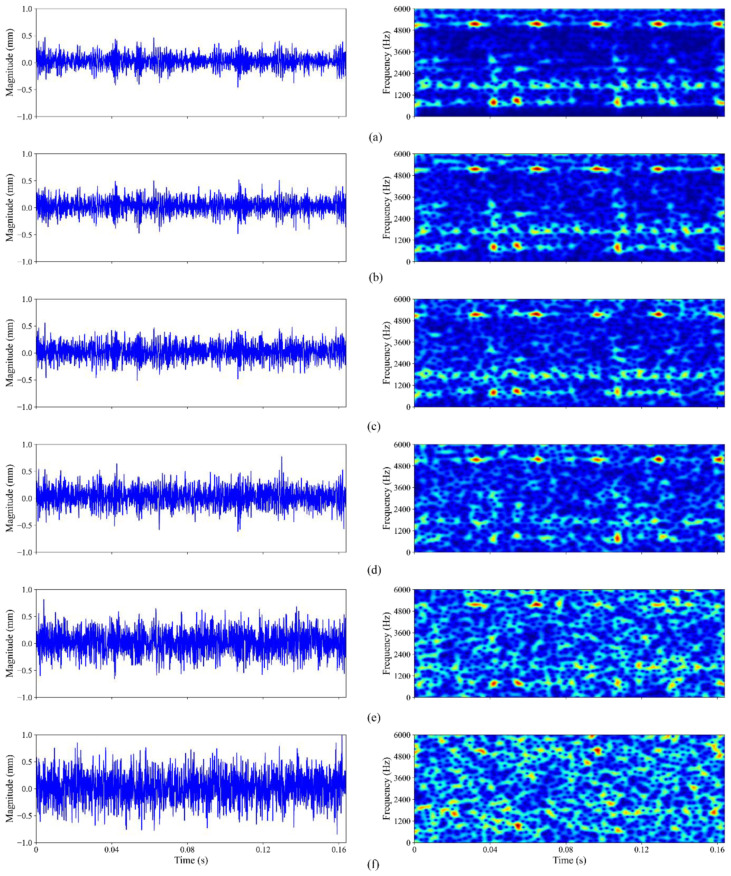
Time-frequency image of bearing signals with (**a**) no noise and added Gaussian noise of (**b**) 6 dB; (**c**) 3 dB; (**d**) 0 dB; (**e**) −3 dB; (**f**) −6 dB.

**Figure 9 sensors-23-05607-f009:**
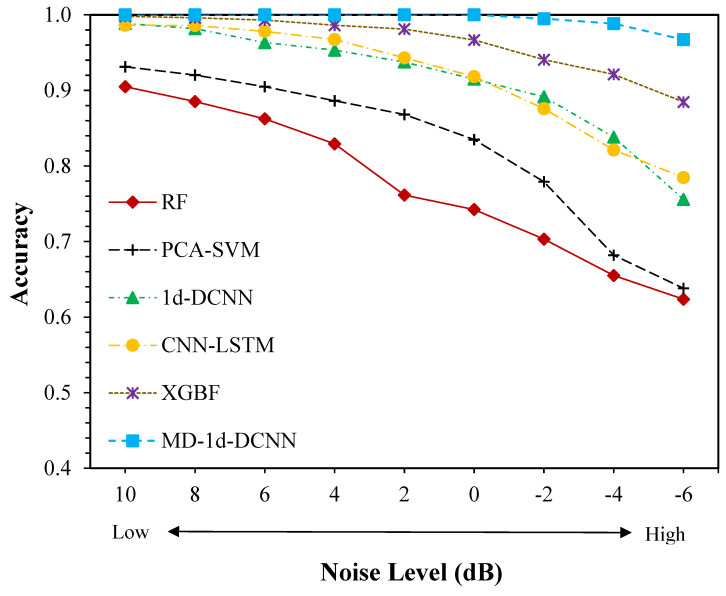
Comparison results of different diagnostic methods on CWRU dataset with different noise levels.

**Figure 10 sensors-23-05607-f010:**
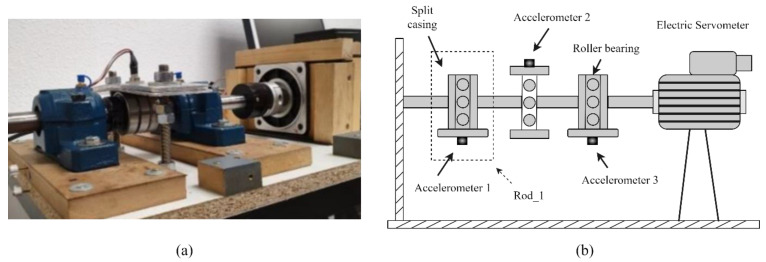
(**a**) The CITEF bearing fault test rig; (**b**) Schematic diagram of the CITEF experimental setup.

**Figure 11 sensors-23-05607-f011:**
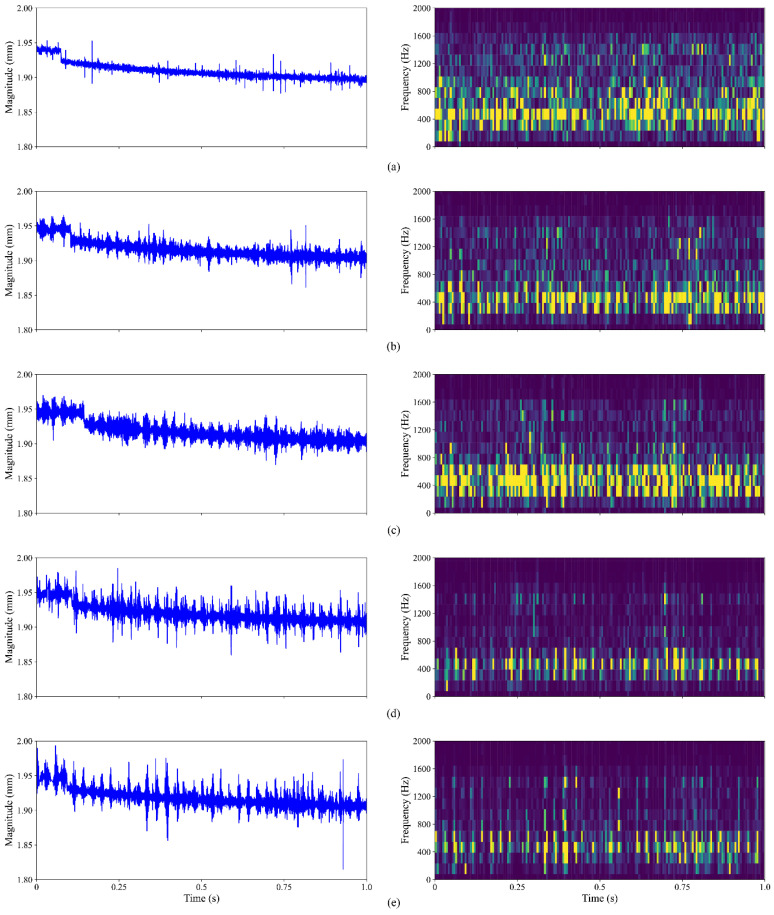
Time-frequency spectrograms of raw signal of combined fault class (**a**) 0; (**b**) 1; (**c**) 2; (**d**) 3; (**e**) 4.

**Figure 12 sensors-23-05607-f012:**
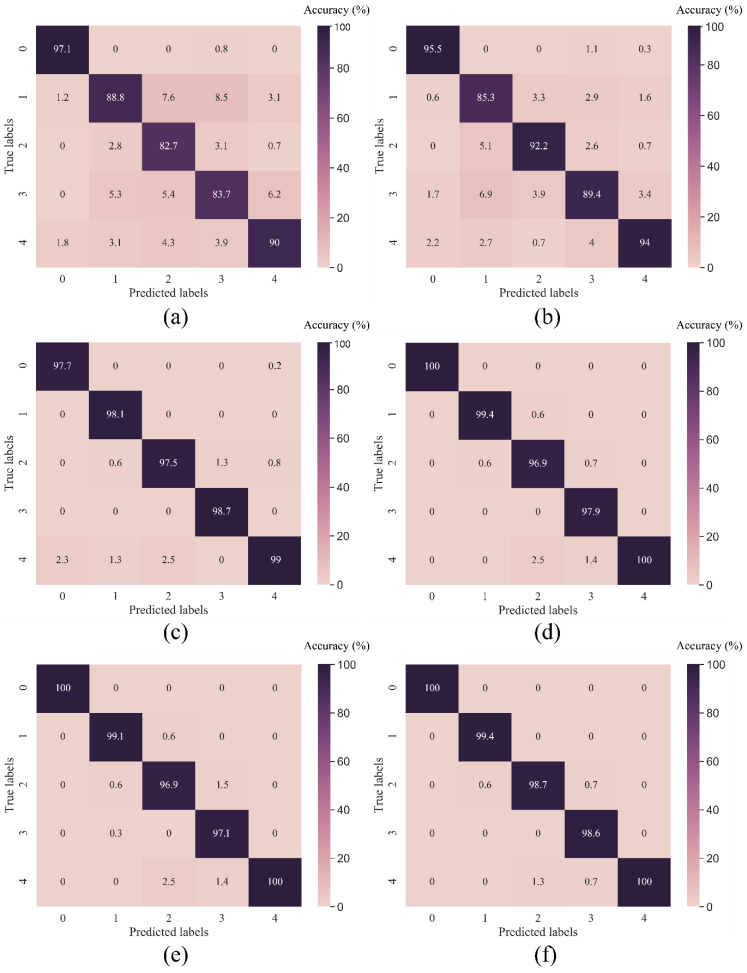
Confusion matrices of various methods in testing samples of CITEF dataset (**a**) RF; (**b**) PCA-SVM; (**c**) 1d-DCNN; (**d**) CNN-LSTM; (**e**) XGBF; (**f**) MD-1d-DCNN.

**Figure 13 sensors-23-05607-f013:**
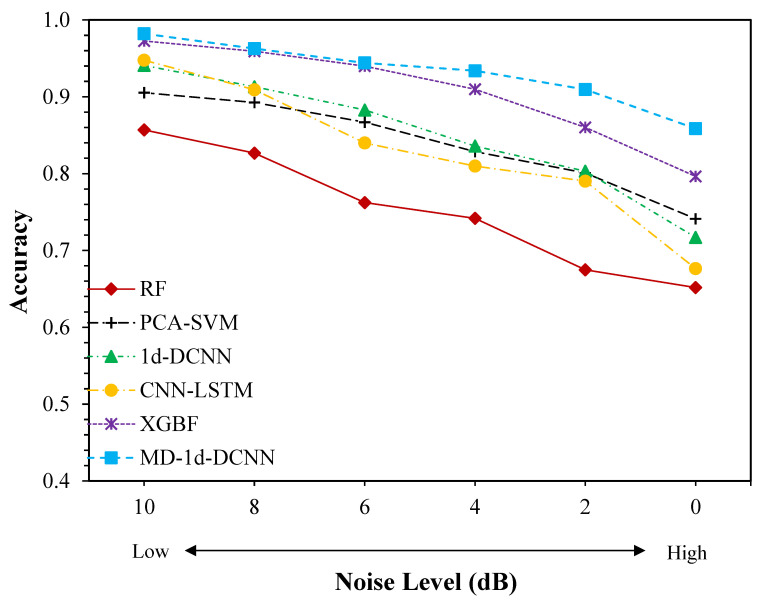
Comparison results of different diagnostic methods on CITEF dataset with different noise levels.

**Table 1 sensors-23-05607-t001:** Mathematical expressions of statistical features from different domains.

Time Domain		Frequency Domain		Time-Frequency Domain	
Mean-absolute	$p_{1}^{t}=\frac{1}{N}\overset{N}{\underset{n=1}{\sum }}\left|x\left(n\right)\right|$	Position change indicator	$p_{1}^{f}=\sqrt{\frac{\sum_{k=1}^{K}f_{k}^{2}s\left(k\right)}{\sum_{k=1}^{K}s\left(k\right)}}$	Relative energy	$E_{i}=\frac{E_{i}^{\prime }}{\sum_{i=1}^{n}E_{i}^{\prime }}$ where $E_{i}^{\prime }=$ energy of each frequency band, $\overset{n}{\underset{i=1}{\sum }}E_{i}$=1
Root-mean-square	$p_{2}^{t}=\sqrt{\frac{\sum_{n=1}^{N}\left(x\left(n\right)\right){)}^{2}}{N}}$		$p_{2}^{f}=\sqrt{\frac{\sum_{k=1}^{K}f_{k}^{4}s\left(k\right)}{\sum_{k=1}^{K}f_{k}^{2}s\left(k\right)}}$		
Square-mean-root	$p_{3}^{t}={\left(\frac{\sum_{n=1}^{N}\sqrt{\left|x\left(n\right)\right|}}{N}\right)}^{2}$		$p_{3}^{f}=\frac{\sum_{k=1}^{K}f_{k}s\left(k\right)}{\sum_{k=1}^{K}s\left(k\right)}$		
Peak-to-peak	$p_{4}^{t}=\max\left(x\right)-\min\left(x\right)$	Energy indicator	$p_{4}^{f}=\frac{1}{K}\overset{K}{\underset{k=1}{\sum }}\left|s\left(k\right)\right|$		
Kurtosis	$p_{5}^{t}=\frac{\frac{1}{N}\sum_{n=1}^{N}{\left(x\left(n\right)\right)}^{4}}{{\left(p_{2}^{t}\right)}^{4}}$		$p_{5}^{f}=\frac{\sum_{k=1}^{K}{\left(s\left(k\right)-p_{4}^{f}\right)}^{2}}{K}$	Energy entropy	$H=-\overset{n}{\underset{i=1}{\sum }}\frac{E_{i}}{E}\log\frac{E_{i}}{E}$ where $E=\overset{n}{\underset{i=1}{\sum }}E_{i}$
Crest factor	$p_{6}^{t}=\frac{\max\left(\left|x\right|\right)}{p_{2}^{t}}$		$p_{6}^{f}=\max\left(s\right)$		
Shape factor	$p_{7}^{t}=\frac{p_{2}^{t}}{p_{1}^{t}}$		$p_{7}^{f}=\sqrt{\frac{\sum_{k=1}^{K}{\left(f_{k}-p_{3}^{f}\right)}^{2}s\left(k\right)}{K}}$		
Impulse	$p_{8}^{t}=\frac{\max\left(\left|x\right|\right)}{p_{1}^{t}}$		$p_{8}^{f}=\frac{\sum_{k=1}^{K}{\left(f_{k}-p_{3}^{f}\right)}^{3}s\left(k\right)}{K{\left(p_{7}^{f}\right)}^{3}}$		

**Table 2 sensors-23-05607-t002:** Specific parameters of the proposed model architecture.

Operation	Layer	Parameter
1d-DCNN feature extraction	1d-DCNN (LeakyReLU, BN, MaxPooling)	Filter: 16, kernel: 64, dilation rate: 1, pool size: 2
	1d-DCNN (LeakyReLU, BN, MaxPooling)	Filter: 32, kernel: 16, dilation rate: 2, pool size: 2
	1d-DCNN (LeakyReLU, BN, MaxPooling)	Filter: 32, kernel: 16, dilation rate: 4, pool size: 2
	1d-DCNN (LeakyReLU, BN, MaxPooling)	Filter: 64, kernel: 16, dilation rate: 4, pool size: 2
	Dense	Units: 16
	Dropout	Dropout rate: 0.2
Multi-domain feature extraction	Dense	Units: 100
	Dense	Units: 50
	Dense	Units: 16
	Dropout	Dropout Rate: 0.5
Output layer	Dense	Units: 32
	Dense	Units: 12

**Table 3 sensors-23-05607-t003:** Description of the CWRU rolling bearing dataset.

Fault Type	Fault Diameter (in)	Class Label	Sample Size
Normal (N)	-	0	120
Inner race (IR)	0.007	1	120
	0.014	2	120
	0.021	3	120
	0.028	4	120
Rolling element (RA)	0.007	5	120
	0.014	6	120
	0.021	7	120
	0.028	8	120
Outer race (OR)	0.007	9	120
	0.014	10	120
	0.021	11	120

**Table 4 sensors-23-05607-t004:** Comparison of accuracy with other models under a noise-free environment for CWRU dataset.

	Methods					
	MD-1d-DCNN	XGBF	CNN-LSTM	1d-DCNN	RF	PCA-SVM
Accuracy	**100.0%**	99.70%	99.89%	99.89%	91.94%	95.32%
Time	**55 s**	133 s	101 s	36 s	96 s	71 s

**Table 5 sensors-23-05607-t005:** Description of the CITEF rolling bearing dataset.

Fault Type	Damage Description			Fault Class	Sample Size
	Location	Area (mm^2^)	Depth (mm)		
Rolling element (RE) & Outer race (OR)	RE OR	0 0	0 0	0	765
	RE OR	11.05 25.874	0.006 0.007	1	765
	RE OR	11.57 28.928	0.014 0.013	2	765
	RE OR	11.7 31.983	0.019 0.02	3	765
	RE OR	13 33.241	0.027 0.028	4	765

**Table 6 sensors-23-05607-t006:** Comparison of accuracy with other models under a noise-free environment for CITEF dataset.

	Methods					
	MD-1d-DCNN	XGBF	CNN-LSTM	1d-DCNN	RF	PCA-SVM
Accuracy	**99.35%**	98.85%	98.95%	98.04%	88.27%	91.36%
Time	**147 s**	546 s	413 s	56 s	515 s	394 s

## Data Availability

All datesets used are available online. CWRU bearing dataset: https://engineering.case.edu/bearingdatacenter/download-data-file; CITEF bearing dataset: https://zenodo.org/record/3898942.

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
