# Peer review of "Intelligent Bearing Fault Diagnosis Based on Feature Fusion of One-Dimensional Dilated CNN and Multi-Domain Signal Processing"

_sensors, 2023, doi:10.3390/s23125607_

Round 1

Reviewer 1 Report

Formal comments:

on Figure 1 - missing descriptions on the axes on the left - Time domain and Frequency domain

on Figure 8 and 11 - what do the colored points in the Time - frequency spectra represent - missing colormap label

Content comments:

In real operations, the nature of frequency spectra changes over time for rolling bearings - usually U-shaped. Is it possible with the presented methodology to track and detect damage to rolling bearings that at which stage of damage it is?

Can the presented methodology distinguish other fault frequencies of the whole machine, such as rotational frequency, fault frequencies of gears, clutches, etc.?

Is it possible to apply the results and evaluate the machines on the basis of ISO 20816?

Reviewer 2 Report

General Comments

 The manuscript entitled “Intelligent Bearing Fault Diagnosis Based on Feature Fusion of One-Dimensional Dilated CNN and Multi-Domain Signal Processing” is relatively well written, with just some typological and grammatical errors, needing a quick English revision. The theme is relevant, the idea to propose new model for bearing fault diagnosis is interesting and the application meets the current needs of improvement in this field.

As the paper has to be evaluated under the point of view of the results presented and the engineering relevance, there is a need for improvements in the paper before it can be accepted for publication. In the specific comments, I bring some points to be included and/or analyzed more carefully by the authors, in order to make the manuscript suitable for publication and to bring greater contributions to the scientific community.

 Specific comments and questions

1) The last paragraph of introduction is not necessary for a paper (the part specifying the paper structure).

2) The paper only compares the different methods in terms of accuracy, however, for application in real machines, the total processing time spent by each method is extremely important. Authors should include this information.

3) The authors argue that the main advantage of the proposed method would be the application in "real-world situations". However, they only make comparisons with databases from laboratory test rigs. The artificial introduction of gaussian white-noise does not make the data similar to real machines in operation. The authors suggest these types of tests as future developments. Perhaps it would be interesting to have these results to justify the novelty of the method and its real applicability.

4) Currently, the search for increasingly simpler methods that show robustness and good accuracy is being more valued than methods that have extreme accuracy but require a complex signal processing and costly processing time. How do the authors see the viability of the proposed algorithm in this scenario?

5) Finally, the authors should discuss the limitations and drawbacks of the proposed model, from a practical application point of view. Just by reading the paper it is already possible to see some.

As already mentioned, the paper needs a quick revision of the used English.

Round 2

Reviewer 2 Report

Of the five points I raised during the review, the authors only included the first and last in the paper. The other three were superficially answered in the letter, however, nothing was added in the paper. I still believe it is important for the authors to do what was asked in the other items, or at least better discuss these points in the paper.
